# Improving Generalization with Approximate Factored Value Functions

## Abstract

Reinforcement learning in general unstructured MDPs presents a challenging learning problem. However, certain kinds of MDP structures, such as factorization, are known to make the problem simpler. This fact is often not useful in more complex tasks because complex MDPs with high-dimensional state spaces do not often exhibit such structure, and even if they do, the structure itself is typically unknown. In this work, we instead turn this observation on its head: instead of developing algorithms for structured MDPs, we propose a representation learning algorithm that approximates an unstructured MDP with one that has factorized structure. We then use these factors as a more convenient state representation for downstream learning. The particular structure that we leverage is reward factorization, which defines a more compact class of MDPs that admit factorized value functions. We show that our proposed approach, **A**pproximately **Fa**ctored **R**epresentations (AFaR), can be easily combined with existing RL algorithms, leading to faster training (better sample complexity) and robust zero-shot transfer (better generalization) on the Procgen benchmark. An interesting future work would be to extend AFaR to learn *factorized* policies that can act on the individual factors that may lead to benefits like better exploration. We empirically verify the effectiveness of our approach in terms of better sample complexity and improved generalization on the ProcGen benchmark and the MiniGrid environments.

## 1 Introduction

Reinforcement Learning problems are often modeled as Markov Decision Processes (MDPs). While the MDP formulation is quite general, it is not always the most optimal. Recent works Zhang et al. (2021); Sodhani et al. (2021a) have proposed introducing additional assumptions to leverage the structure underlying the given task(s). But these approaches may not be useful when working with complex MDPs with high-dimensional state spaces where the structure is often not apparent.

In this work, we propose leveraging structure in the reverse direction. Instead of developing algorithms for structured MDPs, we propose mapping an unstructured MDP to an approximate structured MDP. We exploit the structure for improved sample efficiency and generalization. In the scope of this work, we focus on the *Factored Reward MDP* which has factorized states and reward and provides computational benefit as they are more *compact* than standard MDPs.

We propose a representation learning technique, called **A**pproximately **Fa**ctored **R**epresentations (AFaR), that can map any unstructured MDP into an approximate Factored Reward MDP. We show that Factored Reward MDPs exhibit factored value functions which may allow for better generalization to novel combinations of those factors. Existing RL algorithms can be extended to exploit the structure of the Factored Reward MDP, and we show that this can lead to sample efficiency gains and improved generalization even when that mapping is approximate.

## 2 Preliminaries

A **Markov Decision Process** (MDP) Puterman (1995) is defined by a tuple $\langle \mathcal{S}, \mathcal{A}, R, T, \gamma \rangle$, where $\mathcal{S}$ is the set of states, $\mathcal{A}$ is the set of actions, $R : \mathcal{S} \times \mathcal{A} \to \mathbb{R}$ is the reward function, $T : \mathcal{S} \times \mathcal{A} \to Dist(\mathcal{S})$ is the environment transition probability function, and $\gamma \in [0, 1)$ is the discount factor. The

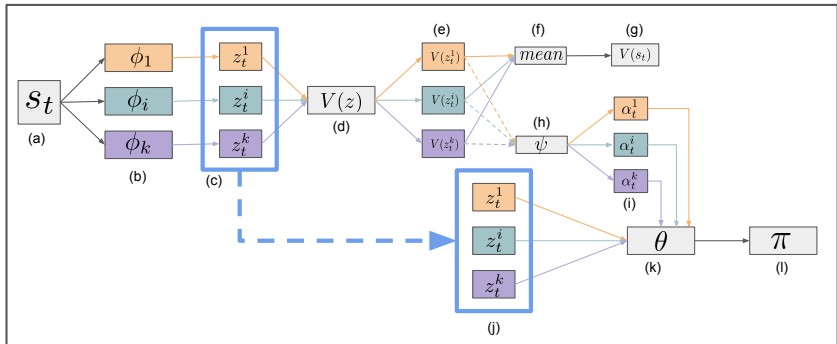

Figure 1: Architecture of the AFaR model: Given an input state $s$ (component $a$), an mixture of $k$ encoders, $\phi_1 \cdots \phi_k$ (component $b$), is used to compute $k$ factor representations, denoted as $z_t^i = \phi_i(s_t) \forall i \in \{1, \cdots k\}$ (component $c$). State value is computed for each factor (components $d$ and $e$) and the overall state value is obtained by averaging over the individual state values (components $f$ and $g$). The state values are also used to compute the attention scores $\alpha$ (components $h$ and $i$) which is used to for aggregating the factor representations(component $k$). Dashed lines indicate that gradient does not flow through those components or computations. The blue box/arrows connecting components $c$ and $j$ show that the factor representations are detached before pass to the feature aggregation module, $\theta$. The aggregated feature representation is used to select the action from the a given policy (component $l$).

value function of policy $\pi$ is defined as: $V_\pi(s) = E_\pi[\sum_{t=0}^{\infty} \gamma^t R(s_{t+1})|S_0 = s]$. The optimal value function $V^*$ is the maximum value function over the class of stationary policies.

A **Factored Reward Markov Decision Process** (Factored Reward MDP) is an MDP where the state and the reward can be factored into variables. For example, a state $s_t$ can be factored into $k$ state factors $s_t^1, \cdots, s_t^k$. We rely on the assumption that for given full states $s_t, s_{t+1} \in \mathcal{S}$, action $a_t \in \mathcal{A}$, and $s_t^i$ denoting the $i^{\text{th}}$ factor of the state $s_t$, we have $R(s_{t+1}|s_t, a_t) = \sum_i R(s_{t+1}^i|s_t, a_t)$.

It can be shown that the Factored Reward MDPs emit factored value functions i.e. the state-action value functions for any policy $\pi$ can be factorized as: $V^\pi(s_t) = \sum_{i=1}^k V_i^\pi(s_t^i)$ and $Q^\pi(s_t, a_t) = \sum_{i=1}^k Q_i^\pi(z_t^i, a_t)$, where $V_i^\pi(s_t^i) = \sum_{t'=t}^{\infty} \gamma^{t'} \sum_{s_{t'+1}} T(s_{t'+1}|s_{t'}, a_{t'}) r(s_{t'}^i, a_{t'}))$ and $Q_i^\pi(s_t^i, a_t) = r(s_t^i, a_t) + \sum_{t'=t+1}^{\infty} \gamma^{t'} \sum_{s_{t'+1}} T(s_{t'+1}|s_{t'}, a_{t'}) r(s_{t'}^i, a_{t'}))$ are the state value and the state-action value functions for the $i^{th}$ factor, respectively.

# 3 AFaR: Learning Approximate Factored Representations

In Section 2, we noted that Factored Reward MDPs emit factored value functions. We now reverse this observation to propose a representation learning technique that induces such factorization. Specifically, we approximate the value of a state as the sum of the values corresponding to the different factors. These factors are learnt using Factor Encoders (denoted as $\phi$) and are described in Section 3.1. This factorization can be seen as applying the *mean-field principle* to approximate the value function of a state in terms of the value function of the factors, similar to Peyrard & Sabbadin (2006). We refer to our proposed approach as **A**pproximately **Fa**ctored **R**epresentations, or AFaR. Next, we discuss how to leverage this form of structured representation for downstream control.

## 3.1 Factor Encoders

The system comprises $k$ encoders, denoted as $\phi_1, \cdots, \phi_k$, corresponding to $k$ different factors. Here, $k$ is a hyper-parameter. These encoders learn the factor representations, which are denoted as $z_t^i = \phi_i(s_t), \forall i \in \{1, \cdots, k\}$. For each factor encoder $\phi_i$, we compute the corresponding state value. The overall state value is obtained by averaging over the state values corresponding to the individual factors. The value functions and encoders are trained using the critic loss. We note that the value functions are shared across all factors.

## 4 EXTRACTING A POLICY FROM FACTORIZED Q-FUNCTIONS

Previous works have shown that factorized state and value functions need not lead to factorized policies Liberatore (2002). As such, we need to combine the factorized representation into a single representation that can be fed into a universal policy (shared across the factors). The learning agent aggregates the representations using an *Aggregation Module*, denoted by $\theta$, that selects one (or more) factors to attend over at every timestep. The selection mechanism enables the policy to condition only on the important factors and ignore the irrelevant factors. For example, if the agent is searching for a key to open a door, it may not have to attend to the door till it finds the key.

### 4.1 ATTENDING TO THE FACTORIZED REPRESENTATIONS

Given a set of factor representations $z_t^1, \cdots, z_t^k$, we want to learn attention scores $\alpha_t^1, \cdots, \alpha_t^k$ (represented jointly as $\alpha_t$) that correspond to the relative importance of the factors. This would enable the policy to attend to the most relevant factors. Note that the output of the state value function $V$ (trained as a component of actor-critic algorithms) already captures the *value* of a given factor in terms of the expected returns. Comparing the state-value functions for two states can approximate what state is expected to lead to higher returns for the learning agent. Extending this argument to the factors, a factor with a higher value of the state value function will lead to a higher expected return than a factor with a lower value. Motivated by this insight, we use the state value for the factors as a proxy for computing the relative importance of the factors. Specifically, we compute the attention scores as: $\alpha_t = \psi(V(z_t^1), \cdots, V(z_t^k))$, where $\psi$ represents the *attention module* (instantiated using the softmax operation). The state values are *detached* from the computation graph before feeding to the *attention module* to ensure that only the critic loss updates the value function.

### 4.2 AGGREGATION MODULE

Given the factor representations, $z_t^1 \cdots z_t^k$, and attention weights, $a_t^1 \cdots a_t^k$, we consider the following operations: (i) *soft-attention* where the factor representations are first weighted using the attention values $\alpha$ and are then averaged to obtain an aggregated representation, and (ii) *sparse, top-m attention* where we select the factor representations corresponding to the top-$m$ values from $\alpha$. These selected representations are averaged using the attention values to compute the aggregated representation. Here $m$ is a hyper-parameter. For the case of *soft-attention*, the aggregated output can be computed as $\theta(z_t^1, \cdots, z_t^k, a_t^1, \cdots, a_t^k,) = \sum_{i=1}^{k} z_t^i \times \alpha_t^i$. For the case of *sparse, top-m attention*, the aggregated output can be computed as $\theta(z_t^1, \cdots, z_t^k, a_t^1, \cdots, a_t^k, m) = \sum_{i=1}^{k} z_t^i \times \alpha_t^i \times \mathbb{1}_{\alpha_{t,m}^i}$, where $\mathbb{1}_{\alpha_{t,m}^i}$ is an indicator variable set to 1 if $\alpha_t^i$ is among the largest $m$ values and 0 otherwise.

## 5 EXPERIMENTS

We use the Procgen benchmark Cobbe et al. (2020) and the MiniGrid Environments Chevalier-Boisvert et al. (2018) for verifying the usefulness of the AFaR technique. The Procgen benchmark comprises 16 procedurally generated environments, each representing a distribution of levels. We train the agent on a fixed set of levels while testing on the full distribution of levels (generated by sampling levels at random). MiniGrid is a grid-world environment where the agent has to find a key, unlock a door and pickup an object. For MiniGrid, we use the `MiniGrid-KeyCorridorSxRy` environments where `x` denotes the size of a room and `y` denotes the number of rows. The agent is trained on the `MiniGrid-KeyCorridorS3R3` environment and evaluated on the `MiniGrid-KeyCorridorS3R2` environment in a zero-shot manner.

Since AFaR is a representation learning technique, we need to combine it with some policy algorithm. For MiniGrid environments, we use Rewarding Impact-Driven Exploration (RIDE) Raileanu & Rocktäschel (2020), and for Procgen environments, we use Data-regularized Actor-Critic (DrAC) Raileanu et al. (2020). Demonstrating that AFaR can improve the performance of both baselines shows that AFaR is useful for a variety of actor-critic algorithms. We do not change any hyper-parameters when training the adapted model, showing that AFaR can be used with existing baselines without tuning all the hyper-parameters from scratch. We run all experiments with 10 seeds and report mean and standard error.

| Environment | DrAC | AFaR |
|---|---|---|
| Bigfish | $8.88 \pm 0.89$ | $\mathbf{12.53 \pm 0.53}*$ |
| Bossfight | $7.77 \pm 0.21$ | $\mathbf{7.79 \pm 0.21}$ |
| Caveflyer | $4.37 \pm 0.21$ | $\mathbf{5.63 \pm 0.25}*$ |
| Chaser | $6.56 \pm 0.24$ | $\mathbf{7.07 \pm 0.22}$ |
| Climber | $6.76 \pm 0.14$ | $\mathbf{7.14 \pm 0.1}*$ |
| Coinrun | $8.62 \pm 0.06$ | $\mathbf{8.62 \pm 0.09}$ |
| Dodgeball | $4.78 \pm 0.23$ | $\mathbf{5.24 \pm 0.18}$ |
| Fruitbot | $27.85 \pm 0.28$ | $\mathbf{28.22 \pm 0.15}$ |
| Heist | $3.96 \pm 0.12$ | $\mathbf{4.93 \pm 0.15}*$ |
| Jumper | $\mathbf{5.8 \pm 0.11}$ | $5.58 \pm 0.09$ |
| Leaper | $4.0 \pm 0.3$ | $\mathbf{5.19 \pm 0.33}*$ |
| Maze | $6.33 \pm 0.11$ | $\mathbf{6.72 \pm 0.07}*$ |
| Miner | $9.63 \pm 0.13$ | $\mathbf{10.11 \pm 0.09}*$ |
| Ninja | $5.41 \pm 0.13$ | $\mathbf{5.49 \pm 0.11}$ |
| Plunder | $8.12 \pm 0.48$ | $\mathbf{9.27 \pm 0.34}$ |
| Starpilot | $\mathbf{29.67 \pm 0.85}$ | $28.89 \pm 0.65$ |

Figure 2: In the table, we compare the performance of AFaR with DrAC (baseline approach) in terms of the score (on the evaluation environments) at the end of training when training with 200 levels. We report the mean and the standard error over 10 runs. The results marked in **bold** are the best performances for each environment and the results marked with **\*** are denote statistically significant results (Welch's $t$-test, with the significance level ($p$) is set to 0.05). The proposed approach improves over the baseline in 13 (out of 16) environments, with statistically significant improvement in 7 environments. In the plots, we compare the performance of AFaR with RIDE (baseline approach) on `MiniGrid-KeyCorridorS3R3-v0` and `MiniGrid-KeyCorridorS3R2-v0` environments (middle and right frames respectively). We note that the agent was trained on `MiniGrid-KeyCorridorS3R3-v0` and evaluated on `MiniGrid-KeyCorridorS3R2-v0` environment at regular intervals in a zero-shot manner. We also include a modified version of AFaR algorithm where we replace the attention mechanism with the *mean* operation, denoted as AFaR-mean. For both the environments, the AFaR approach improves the sample efficiency of the baseline and sparse attention always outperforms the case of using *mean* operation.

**Results.** In the table in Figure 2, we note that AFaR significantly outperforms the DrAC baseline on 7 environments, while improving the performance on 13 environments. On the MiniGrid environments, AFaR improves the sample efficiency of the RIDE baseline for both the training environment (`MiniGrid-KeyCorridorS3R3-v0`) and the zero-shot evaluation environment (`MiniGrid-KeyCorridorS3R2-v0`).

# 6 RELATED WORK

**Factored MDPs** Boutilier et al. (1995; 1999) are a special class of MDPs where the state, dynamics and reward can be factored into variables. However, factored transitions are a strong assumption and do not necessarily give rise to factored value functions. We operate on Factored Reward MDPs, which admits factorized value functions. Such factorization is related to the work on **Successor Features** Dayan (1993); Barreto et al. (2017) where the value functions are represented as a special class of basis functions. Koller & Parr (1999) is quite close to our work as they also proposed imposing the additive structure on the value function. However, they use hand-designed basis functions while we learn the factorization as part of the training process. Our work is also related to **Mixture of Experts** (MoE) which have been used in the context of multi-task learning Sodhani et al. (2021b), hierarchical reinforcement learning Goyal et al. (2020) and multi-agent learning He & Boyd-Graber (2016). In contrast to these works, we use MoE to map a given MDP to an approximate Factored Reward MDP and train RL agents on the Factored Reward MDP.

# 7 CONCLUSION

In this work, we study a form of structured environment with additively factorized rewards. We call this setup *Factored Reward MDP*. A nice feature of this structured MDP is that the value function also factorizes. We design an algorithm, AFaR, that learns an approximate Factored Reward MDP of any given environment. We show that AFaR can be easily combined with existing RL algorithms, leading to improved sample efficiency and generalization performance in both MiniGrid and Procgen environments. An interesting future work would be to extend AFaR to learn *factorized* policies that can act on the individual factors that may lead to benefits like better exploration.

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
