# OpenReview forum: "Improving Generalization with Approximate Factored Value Functions"
_ICLR.cc/2022/Workshop/OSC — ICLR2022 OSC  Poster_

### Official Review · Reviewer_FuVs · 2022-03-13

**Rating:** 2
**Confidence:** 3

**Review:**

**Summary**
This work proposes a novel factored architecture that aims to leverage the reward factorization structure. The main idea is to learn a mixture of state encoders to compute factor representations. The computed factor representations are used to compute factor value which is aggregated to compute the state value, and also used to compute the policy via attention mechanism. The experiment was conducted on Procgen and Minigrid domains where the architecture of the baseline model is replaced with the proposed architecture.The result shows that the proposed architecture improves the sample efficiency of the baseline method.

**Pros**
* The proposed idea is interesting and well-motivated
* The paper reads well
* The experiment was conducted on diverse tasks

**Cons**
* The performance improvement is marginal
* It is unclear whether it is a fair comparison in terms of the architecture capacity. (see below)
* The presentation can be improved (see below)

**Major comments**
* Is it a fair comparison in terms of the capacity of model architecture? It seems the proposed architecture multiplies the capacity of the model by a number of factors. It should be made sure that both architectures of the baseline and baseline+AFaR should have the same capacity for fair comparison.
* Why disconnect gradients from policy for learning representation? The gradient coming from the policy loss is a major source of learning representation. It would be better if authors provide more intuitive reasoning behind this design choice.
* Although it is a design choice, it may be more natural to use summation instead of mean for computing state-value following the factored reward MDP definition. So it would be better to present a more intuitive justification for such a design choice.
* It would be more interesting if authors can analyze the different “mode” of policy learned for each factor.


**Minor comments**
* The experiment result for “sparse” is missing in Figure 2, but “sparse” is mentioned in the comment of Figure 2 and section 4.2.

---

### Official Review · Reviewer_R8U7 · 2022-03-15

**Rating:** 2
**Confidence:** 3

**Review:**

The paper proposes to learn state factorization with the assumption that the MDP reward is a sum of rewards for individual factors. The proposed model uses k encoders to learn k individual factors and an attention mechanism to condense the k factors into a single input to a policy. Predicting attention weights based on individual factor values is an interesting design decision.

Strengths: The paper proposes novel approach that is evaluated on the challenging procgen benchmark, which tests generalization.

Weaknesses: There is no analysis of what the individual factors learn. I'm especially curious if the model chooses to use all factors. Additionally, the number of trainable parameters in the proposed method and the baseline should be compared.

This is a relevant and interesting paper for the workshop.

---

### Decision · Program_Chairs · 2022-03-24

**Decision:**

Accept (Poster)

**Comment:**

The reviewers agree the paper should be accepted at the workshop. Congratulations!

The authors are encouraged to take the points raised by reviewers into account when preparing the camera-ready version.